# Facilitators and Barriers Associated with Uptake of HIV Self-Testing among Men Who Have Sex with Men in Chongqing, China: A Cross-Sectional Survey

**DOI:** 10.3390/ijerph17051634

**Published:** 2020-03-03

**Authors:** Ying Liu, Guohui Wu, Rongrong Lu, Rong Ou, Ling Hu, Yiping Yin, Yangchang Zhang, Hongjia Yan, Yong Zhao, Yetao Luo, Mengliang Ye

**Affiliations:** 1Department of Epidemiology and Health Statistics, School of Public Health and Management, Chongqing Medical University, Chongqing 400016, China; liuying@stu.cqmu.edu.cn (Y.L.); 2018111007@stu.cqmu.edu.cn (L.H.); 2014222546@stu.cqmu.cn (Y.Z.); 2016111018@stu.cqmu.edu.cn (Y.L.); 2The Second Clinical College, Chongqing Medical University, Chongqing 400016, China; 3Institute for AIDS/STD Control and Prevention, Chongqing Center for Disease Control and Prevention, Chongqing 400042, China; wgh68803652@163.com (G.W.); lurrong@163.com (R.L.); 4Department of Medical Informatics, Library, Chongqing Medical University, Chongqing 400016, China; ourong@cqmu.edu.cn; 5Mechanical Innovation Center for Social Risk Governance in Health, School of Public Health and Management, Chongqing Medical University, Chongqing 400016, China; 2017222834@stu.cqmu.edu.cn (Y.Y.); 2016223498@stu.cqmu.edu.cn (H.Y.); zhaoyong@cqmu.edu.cn (Y.Z.)

**Keywords:** men who have sex with men (MSM), HIV self-testing (HIVST), accuracy, factors

## Abstract

While studies on human immunodeficiency virus self-testing (HIVST) continue to accumulate after the World Health Organization’s recommendation of HIVST as an additional approach to HIV testing services in 2016, few studies have focused on men who have sex with men (MSM) in Chinese cities. A cross-sectional study was conducted to describe the HIVST status of MSM in Chongqing, China. MSM participants were recruited by random sampling, and qualified interviewers collected data, using confidential self-administered questionnaires. Blood specimens were collected for HIV antibody detection. The survey evaluated the uptake and accuracy of HIVST kits and identified factors that may be associated with HIVST. The proportion of HIVST uptake was 15.6%. The sensitivity and specificity of HIVST were 74.2% (95% confidence interval [CI] 66.6%–80.7%) and 99.0% (95% CI 96.9%–99.7%), respectively. The consistency between the HIVST kit and antibody detection results was 90.5% (95% CI 87.5%–93.0%), and the Kappa value was 0.777 (*p* < 0.001). The positive predictive value of self-testing kits is 80.9% and the negative predictive value is 17.7%. Having been tested ≥2 times in the last year, higher educational levels, and higher scores of basic HIV/AIDS knowledge facilitated higher uptake of HIVST. Self-reported existing barriers for HIVST uptake included older age, marital status, and having resided in Chongqing for more than two years.

## 1. Introduction

Despite global efforts to control human immunodeficiency virus (HIV) among crucial populations, new infections have continuously increased among men who have sex with men (MSM), among whom, the highest incidence, prevalence, and percentage rates of undiagnosed HIV infections are often detected [1,2]. Globally, MSM are 19 times more likely to live with HIV compared with the general population [3]. In 2014, among newly diagnosed cases in China, 25.8% were MSM; this figure is a substantial increase from the rate of 2.5%, reported in 2006 [4]. The HIV prevalence among Chinese MSM in 2015 was approximately 8%; those living in Southwest China showed an even higher prevalence [5]. In 2015, a study showed an overall HIV antibody positive rate of 21.21% among MSM in Chongqing and Sichuan, two typical representative regions of Southwest China [6].

Testing is a vital strategy to prevent and control the global HIV epidemic. Indeed, HIV testing is indispensable for screening and access to HIV care, including counseling on risk reduction and diagnosis [7]. The first goal of the Joint United Nations Program on HIV/AIDS (UNAIDS) 90–90–90 targets is to diagnose 90% of all people living with HIV (PLHIV) by the year 2020 [8]. However, approximately 43% of the PLHIV worldwide remained unaware of their infection status in 2015 [9]. An estimated 850,000 PLHIV cases were reported in China in 2015, but approximately 35% of these cases remained unidentified [10]. Access to testing remains an issue in many regions, and fear of stigma, lack of privacy, and discrimination continue to threaten the acceptance of HIV testing services [11]. HIV testing in China is usually performed by trained medical personnel and is, in most cases, completely facility-based; testing is offered primarily at volunteer counseling testing (VCT) sites, hospitals, specialized HIV/AIDS clinics, and at Centers for Disease Control and Prevention (CDC) offices [12]. A study found that only 28% of MSM are willing to accept free HIV testing from the government program at their local CDC [13]. MSM worry about privacy and confidentiality, fear of stigma, and discrimination related to HIV [13,14,15].

HIV self-testing (HIVST), as a widely accepted and easily accessible HIV screening method, is needed to rapidly increase the uptake of HIV testing services and rates of diagnosis [16,17,18]. The World Health Organization recommended HIVST as an additional approach to HIV testing services in 2016 [19]. The feasibility and acceptability of self-testing have been demonstrated in multiple published studies [20,21,22], including recent reports from China [23,24] and the world [25,26]. Evidence from a systematic literature review [27] suggests that HIVST is globally acceptable to MSM in both high- and low-income settings. Although HIVST is feasible without prior training or supervision, invalid test results and testing errors have been recorded in previous studies [28,29]. More studies are need to improve the development and ensure the quality of self-test kits and broad-scale implementation of HIV self-testing programs for MSM. In China, MSM can obtain HIVST kits through e-commerce websites, HIV/AIDS clinics, hospitals, pharmacies, and Chinese CDC offices [15,30]. Although HIVST kits are widely available and readily accessible in China, only 6.1%–26.2% of MSM use these kits [20,24]. An understanding of the factors that facilitate HIVST is the first step toward eliminating barriers to test uptake. Therefore, the primary aim of this study is to examine HIVST uptake and test-kit accuracy among MSM in Chongqing, China, as well as facilitators and barriers associated with self-test use.

## 2. Materials and Methods

### 2.1. Ethics Statement

This study was reviewed and approved by the Medical Ethics Committee of Chongqing Medical University and the review board of the Chongqing CDC (IRB No. 2017016). All participants signed informed consent forms before responding to the questions. Participation was voluntary, and participants could choose not to complete part of, or all of, the questionnaire.

### 2.2. Study Design and Participants

We performed a cross-sectional study, using questionnaires in Chongqing from January to December in 2017. The method was random sampling, through which Chongqing CDC staff carried out a series of HIV/AIDS information sessions in MSM venues, such as bars, bathhouses, clubs, saunas, massage parlors, and parks, and encouraged participation. Figure 1 involves a flow diagram describing the study design and the development of the final study population.

In our study, for the HIVST, the MSM collected his own specimen (blood), performed an HIV test kit, and then interpreted the result, often in a private setting, either alone or with someone he trusts. MSM obtained HIVST kits through e-commerce websites, HIV/AIDS clinics, hospitals, pharmacies, and Chinese CDC offices. However, these products are not regulated by the State Food and Drug Administration (SFDA). The inclusion criteria for all participants were as follws: (1) age ≥ 18 years old; (2) biologically male; and (3) reported anal or oral sex with another male at least once during their lifetime. The exclusion criteria were as follows: (1) with a known HIV-positive status; (2) respondents had never heard of HIV-self testing. Upon completion of the survey, condoms and household material items such as towels were distributed to participants. 

### 2.3. Measures

The questionnaire was developed by research analysts from China CDC, according to the “Operational Manual for the Implementation Program of National AIDS Sentinel Surveillance”. The first part of the questionnaire collected demographic data, including age (years), marital status (unmarried/married), ethnicity (Han/minority), household registration (the present locality/other provinces), length of residence in Chongqing (years), highest level of education (junior high school or below/high school, technical secondary school/college or above), and sexual orientation (homosexual/heterosexual/bisexual/undetermined). In the second part of the questionnaire, a total of eight questions answerable by “yes”, “no”, or “do not know” assessed the participants’ basic knowledge of HIV (e.g., “Is AIDS an incurable serious infectious disease?”). Only correct responses were awarded 1 point; incorrect and “do not know” responses were not awarded any point (Cronbach’s alpha = 0.775; score range 0–8) (Table 1). Respondents with 0–5 points and 6–8 points were categorized with low scores and high scores, respectively, in basic HIV/AIDS knowledge.

The third part of the questionnaire measured HIV-related high-risk factors and HIV testing status in Chongqing. High-risk factors were as follows: (1) “How often do you use condoms when you have sex with a male partner?” (2) “How many regular male partners have you had in the last 6 months?” (3) “Have you ever taken drugs?” (4) “Have you ever been diagnosed with STDs?” Here, we asked men to provide their own definition of a regular partner by indicating how many weeks they would date someone before considering him a regular partner. We defined a new partner as someone newly met or not properly known. HIV testing status questions were as follows: (1) “Do you know that the country has a free HIV testing policy?” (2) “Do you know where free HIV testing is provided?” (3) “Have you performed HIVST? If yes, what was the result?” (4) “How many times have you had HIV testing in the past year?”

A volume of 3–5 mL venous blood was collected from each participant for HIV antibody testing. Plasma HIV antibodies were tested by a third-generation enzyme-linked immunosorbent assay (ELISA) for the first screening. For those with HIV-positive results, the same blood samples were retested with a fourth-generation ELISA reagent as reexamination. If both tests were positive, participants were contacted by CDC staff, to draw a new blood sample again for confirmatory testing (Western blot assay, WB).

### 2.4. Data Analysis

Frequencies and percentages were calculated to summarize the distributions of descriptive variables. The uptake was assessed according to the rate of MSM performing HIVST. The accuracy of self-reported HIVST kit results was assessed by comparison with conventional testing, using HIV antibody detection results. The specific approach is to calculate the former’s sensitivity, specificity, and concordance of the paired fourfold table. The sensitivity and specificity were calculated by using the following formulae:Sensitivity = (true positive)/(true positive + false negative)
Specificity = (true negative)/(true negative + false positive)

Binary logistic regression was used to evaluate correlations between variables. Significant sociodemographic variables with *p* < 0.05 during univariate analysis, as well as other potential variables associated with HIVST uptake in this survey, were included in a multivariate logistic regression model. Age, marital status, length of residence in the locality, level of education, sexual orientation, main venues for finding male partners, scores of basic HIV knowledge and high-risk factors, and HIV prevention status were taken as independent variables; HIVST was considered as the dependent variable. Statistical tests included a two-sided test, and statistical significance was considered at *p* < 0.05. Data were analyzed, using SPSS 22.0 (SPSS Inc., Chicago, IL, USA).

## 3. Results

### 3.1. Sociodemographic Characteristics and HIVST Uptake

The baseline characteristics of the participants are reported in Table 2. Participants ranged in age from 18 years to 76 years (mean age, 29.62 ± 8.635 years), and 87.0% were unmarried. Nearly all participants had completed secondary education or higher. Most participants self-identified as homosexual (82.2%) and 16.4% had performed HIV tests ≥2 times in the past year. 

Among the 3017 participants, 470 (15.6%) reported having previously used an HIV self-test. As shown in Table 2, the uptake rate by household registration in Chongqing was 14.3%; residents of other provinces demonstrated a higher proportion (20.5%). A quarter (24.7%) of the participants residing in Chongqing for <1 year reported HIVST-kit use. A higher uptake rate by number of HIV tests in the last year ≥2 times was 39.6%.

### 3.2. Accuracy

While 470 MSMs reported that they had performed HIVST, we only included 463 MSM results in our analyses, as seven MSM did not report their test results (whether the results were positive or negative). The HIV antibody detection results in the table were obtained from the first screening (ELISA), the reexamination (ELISA), and the confirmatory testing (WB). Among 159 participants with positive HIV antibody detection results, three false-positives were found (Sensitivity = 74.2%; 95% CI 66.6%–80.7%). Among 304 participants with negative HIVST results, 41 false-negatives were obtained (Specificity = 99.0%; 95% CI 96.9%–99.7%). The consistency of results between the two methods was 90.5% (95% CI 87.5%–93.0%), and the paired fourfold-table Kappa value was 0.777 (*p* < 0.001) (Table 3). Based on Table 3 and the prevalence of HIV in this sample, using Bayes’ theorem, the positive predictive value of self-testing kits is 80.9%, and the negative predictive value is 17.7%.

### 3.3. Binary Logistic Regression to Identify Factors Influencing HIV Self-Testing

Binary logistic regression (in Table 4) revealed that age, highest level of education, marital status, scores of basic HIV/AIDS knowledge, number of HIV tests conducted in the last year, and duration of residing in Chongqing were significant independent predictors of HIVST uptake among MSM. Respondents who had attended college and above (adjusted odds ratio (aOR) 1.67, 95% CI 1.02–2.75), who had conducted HIV tests ≥2 times in the last year (aOR 3.34, 95% CI 2.62–4.25), and those who had (aOR 1.66, 95% CI 1.06–2.61) showed significantly positive associations with HIVST uptake. Respondents who were of older age (aOR 0.37, 95% CI 0.25–0.55), who were married (aOR 0.64, 95% CI 0.42–0.98), and those who have resided in Chongqing for >2 years (aOR 0.58, 95% CI 0.35–0.95) showed significantly negative associations with HIVST uptake.

## 4. Discussion

In the present study, the rate of HIVST uptake was 15.6%, which is lower than that in prior studies [12,20]. A decrease in HIVST uptake has also been observed in Hong Kong, likely because HIVST kits are not available for purchase over the counter [24]. In Chongqing, MSM have access to HIVST kits through e-commerce websites, hospitals, and Chongqing CDC offices. However, given the limited availability of counseling and self-test information, only a small group of participants are able to perform self-testing. The government should encourage testing organizations to assist persons who are willing to accept the testing service through various means, such as networks, and provide a toll-free number for links to counseling and care [8]. In our survey, the rate of HIVST uptake by household registration in Chongqing was lower than that in other provinces (14.2% vs. 20.1%). This phenomenon may be attributed to the beliefs of Chongqing locals, who consider HIVST unnecessary because they have access to free VCT. Our study reveals a relatively low rate of HIVST uptake among MSM, which may be due to the availability of VCT and the limited access to counseling and self-test information. Therefore, combining community health services with HIVST, including setting up self-testing kiosks in clinics or outreach community sites/vans and providing toll-free numbers for links to counseling and care, is necessary [8].

In this study, MSM who had undergone HIV testing at least twice in the last year revealed a higher uptake of HIVST (39.6%) than those who had not. In logistic regression analysis, those who had conducted at least two HIV tests in the last year had more positive associations with HIVST uptake compared with those who had conducted HIV tests 0–1 time. These results are consistent with a recent meta-analysis that showed that HIVST can improve HIV testing frequencies among MSM by one or more tests every six months [31]. Further studies are required to confirm the association between HIVST and HIV testing frequency. The CDC recommends MSM to be tested at least once a year; if such individuals engage in high-risk behaviors (e.g., unprotected anal sex, multiple sexual partners, and the use of recreational drugs), an HIV test is recommended every 3–6 months [32,33,34]. As a supplementary testing service, HIVST kits could increase testing frequencies by 3.95-fold among delayed testers, who often complain of inconvenient clinic times, concerns/doubts related to HIV services, and the risk of being stigmatized by staff working in site-based service centers [13,35]. Future intervention programs could target the increased uptake of HIVST to improve HIV testing frequencies.

The results revealed a good consistency (90.5%) between HIVST kit results and HIV antibody detection results. In our study, HIVST was associated with high specificity (99%) and lower-than-expected sensitivity (74.4%). Furthermore, the negative predictive value of self-testing kits is very low (17.7%). The cause might be poor quality of the test kits, test operator inexperience, or performing the test during the acute infection stage. A study by Pilcher et al. [36] demonstrated that the sensitivity of testing may be as low as 54% if HIV testing is performed during the acute infection stage. In another study, the potential for reduced test sensitivity was attributed to test-operator inexperience and error, as well as the lack of a counseling component during self-testing [37]. Although the market for HIVST kits is growing rapidly in China, the quality of these kits varies, and little is known regarding their sensitivity and specificity. The most common platform through which our participants purchased test kits was the Internet; however, the quality of such products is unknown. Another reason behind the low sensitivity obtained in this work is related to the potential low quality of kits available in emerging online shops [38]. The Thirteenth Five-Year Plan (2017–2022) emphasizes that China will “explore strategies to promote HIVST by selling kits in pharmacies and online”. The quality of products sold, however, may not be guaranteed. Regulatory systems are not yet available, and not all vendors comply with national safety guidelines. Hence, provision of HIVST information by health authorities and proper regulation of the sale of over-the-counter test kits targeting, both to offline and online customers, are urgently needed [24].

Our findings show that the significant determinants of HIVST uptake among MSM include age, highest level of education, marital status, basic HIV/AIDS knowledge, number of HIV tests conducted in the last year, and duration of residing in Chongqing. Self-reported existing barriers for HIVST uptake included older age and marriage. This finding is similar to that of a previous study conducted in China, showing that the uptake of HIV testing is strongly associated with factors including age and marital status [39]. In our study, HIVST correlated significantly with MSM who had conducted HIV tests ≥2 times in the last year. This finding is consistent with a study by Han et al. [20], who showed that MSM who had performed HIVST in China tend to have HIV tested within 12 months. Our study demonstrated that higher education levels and scores of basic HIV/AIDS knowledge are facilitators to HIVST uptake among MSM; this finding is similar to that of a study conducted by Li et al. [40]. Wang et al. [41] revealed that higher educational levels are associated with higher odds of having undergone HIVST, consistent with our study. Hence, focusing on MSM with older ages and married MSM, and implementing awareness campaigns addressing concerns about the accuracy and safety of HIVST kits may help increase the acceptance and uptake of HIVST among MSM in China. Further studies needed to be conducted to investigate why MSM in China may or may not seek or use HIVST kits. This study presents several limitations. First, it is cross-sectional in nature; thus, we could not explore the temporal relationship between influencing factors and HIVST. In addition, other more-rigorous research methods are needed to confirm the present findings if further exploration and proof of causality are required. Second, some basic questions relevant to HIV/AIDS knowledge in the modern context, such as questions about HIV/AIDS transmission, testing, treatment, and prevention, are missing. Third, sampling among the MSM population is challenging; hence, our data are somewhat limited, as the participants at our study site may not be a true representative of similar risk groups in other cities.

## 5. Conclusions

Our study revealed that HIVST had low uptake, high specificity, and lower-than-expected sensitivity among MSM in Chongqing. Having been HIV tested ≥2 times in the last year, higher educational levels, and higher scores in basic HIV/AIDS knowledge facilitated a higher uptake of HIVST. Self-reported existing barriers for HIVST uptake included older age, marital status, and having resided in Chongqing for more than two years. Thus, the government should formulate guidelines to ensure the quality of self-testing kits made available to these groups and combine community health services with HIVST, to set up self-testing kiosks in clinics or outreach community sites. Furthermore, China’s public-health policymakers must include HIVST in existing programs and implement awareness campaigns addressing concerns about the accuracy and safety of HIVST kits to encourage high-risk groups, such as MSM, to perform HIVST.

## Figures and Tables

**Figure 1 ijerph-17-01634-f001:**
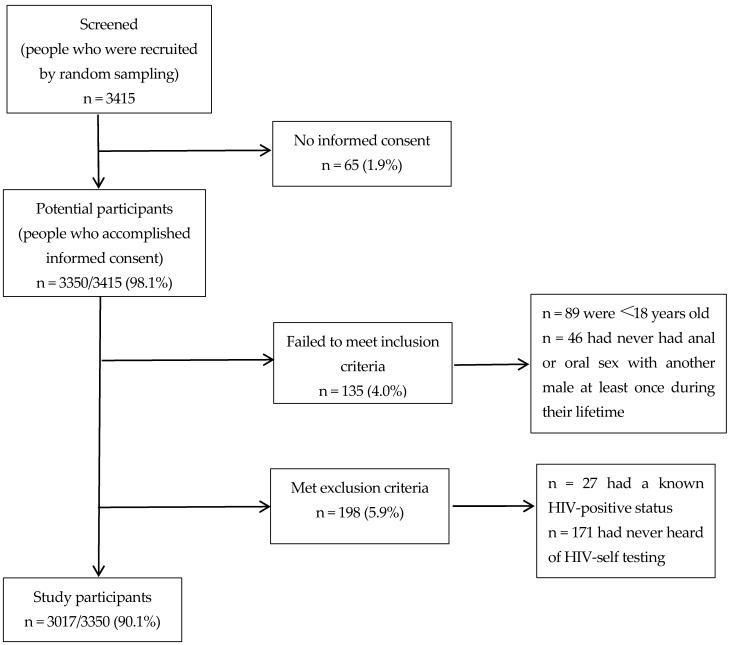
Flow diagram showing the study design, as well as the development of the final study population.

**Table 1 ijerph-17-01634-t001:** Constructs of basic HIV/AIDS knowledge.

Constructs	Items
Basic knowledge	Is AIDS an incurable severe infectious disease?
	Are MSM a group of people who are currently the most severely affected by HIV in China?
	Can you identify a positive patient by their appearance?
	Do sexually transmitted diseases (STDs) increase the risk of HIV infection?
	Does using a condom correctly protect against HIV transmission?
	Does using new drugs (methamphetamine, ecstasy, K powder, and so on) increase the risk of HIV infection?
	Should HIV testing and counseling be actively sought-after high-risk behaviors (needle sharing, drug use, unsafe sex, etc.)?
	Does intentional transmission of HIV/AIDS bear legal responsibility?

**Table 2 ijerph-17-01634-t002:** Characteristics and HIV self-testing (HIVST) uptake of men who have sex with men (MSM) in Chongqing, China (n = 3017).

Characteristics	All Participants	Uptake of HIVST (n = 470)	
	(n = 3017)	n	Percent (95% CI)
**Age**			
18–26 years old	1272	254	20.0% (17.8%–22.3%)
27–36 years old	1283	179	14.0% (12.1%–16.0%)
37–76 years old	462	36	7.8% (5.5%–10.6%)
**Marital status**			
Unmarried	2623	439	16.7% (15.3%–18.2%)
Married	392	30	7.7% (5.2%–10.7%)
**Ethnicity**			
Han	2990	463	15.5% (14.2%–16.8%)
Minority	27	6	22.2% (8.6%–42.3%)
**Household registration**			
Chongqing	2397	342	14.3% (12.9%–15.7%)
Other provinces	620	127	20.5% (17.4%–23.9%)
**Duration of residency in Chongqing**			
<1 year	89	22	24.7% (16.2%–35.0%)
1–2 years	63	11	17.5% (9.1%–29.1%)
>2 years	2865	436	15.2% (13.9%–16.6%)
**Highest level of education**			
Junior high school and below	235	19	8.1% (4.9%–12.3%)
High school or technical secondary school	639	97	15.2% (12.5%–18.2%)
College and above	2142	353	16.5% (14.9%–18.1%)
**Sexual orientation**			
Homosexual	2480	415	16.7% (15.3%–18.3%)
Heterosexual	3	0	
Bisexual	481	45	9.4% (6.9%–12.3%)
Undetermined	52	9	17.3% (8.2%–30.3%)
**Number of HIV tests conducted in the last year**			
0–1 time	1900	321	16.1% (14.5%–17.8%)
≥2 times	374	148	39.6% (34.6%–44.7%)
**Scores of basic HIV/AIDS knowledge**			
0–5	241	23	9.5% (6.1%–14.0%)
6–8	2776	446	16.1% (14.7%–17.5%)

**Table 3 ijerph-17-01634-t003:** Detection results according to type of test (n = 463).

	HIV Antibody Detection Results		Total
	Positive	Negative	
Participant self-testing result			
Positive	118	3	121
Negative	41	301	342
Total	159	304	463

Note: Excluding “I do not know” results; concordance: 90.5% (95% CI 87.5%–93.0%); sensitivity: 74.2% (95% CI 66.6%–80.7%); specificity = 99.0% (95% CI 96.9%–99.7%); Kappa = 0.777(*p* < 0.001).

**Table 4 ijerph-17-01634-t004:** Logistic regression analysis of the factors affecting HIVST among MSM in Chongqing, China.

Variable	Estimated Parameter	*p*-Value	OR ^1^	95.0% CI	aOR ^2^	95.0% CI
**Age**						
18–26 years old			1.00		1.00	
27–36 years old	−0.43	*p* < 0.001 *	0.65	0.53–0.80	0.65	0.53–0.81
≥37 years old	−1.08	*p* < 0.001 *	0.34	0.24–0.49	0.37	0.26–0.55
**Highest level of education**						
Junior high school and below			1.00		1.00	
High school or secondary school	0.71	0.007 *	2.04	1.21–3.41	1.62	0.96–2.75
College and above	0.81	0.001 *	2.24	1.38–3.64	1.67	1.02–2.75
**Marital status**						
Unmarried			1.00		1.00	
Married	−0.89	*p* < 0.001 *	0.41	0.28–0.61	0.64	0.42–0.98
**Ethnicity**						
Han ethnicity			1.00		1.00	
Minor ethnicity	0.44	0.340	1.56	0.63–3.88	1.50	0.60–3.78
**Duration of residing in Chongqing**						
<1 year			1.00		1.00	
1–2 years	−0.44	0.287	0.64	0.29–1.45	0.61	0.27–1.37
>2 years	−0.60	0.016 *	0.55	0.33–0.89	0.58	0.35–0.95
**Scores of basic HIV/AIDS knowledge**						
0–5			1.00		1.00	
6–8	0.60	0.008 *	1.8	1.17–2.82	1.66	1.06–2.61
**Frequency of condom use when having sex with other men**						
Never			1.00		1.00	
Sometimes	0.13	0.657	1.14	0.64–2.03	1.07	0.60–1.93
Every time	−0.25	0.401	0.78	0.44–1.39	0.73	0.41–1.131
**How many regular male sexual partners have you had in the last 6 months?**						
Zero			1.00		1.00	
One	0.16	0.376	1.17	0.83–1.66	1.20	0.84–1.71
Two	0.15	0.456	1.17	0.78–1.74	1.18	0.79–1.778
More than three	0.29	0.365	1.34	0.71–2.52	1.47	0.78–2.80
**Number of HIV tests done conducted in the last year**						
0–1 time			1.00		1.00	
≥2 times	1.23	*p* < 0.001 *	3.41	2.68–4.33	3.34	2.62–4.25
**Have you ever taken drugs?**						
Yes			1.00		1.00	
No	0.280	0.482	1.32	0.61–2.88	1.29	0.59–2.82
**Have you ever been diagnosed with STDs?**						
Yes			1.00		1.00	
No	0.46	0.88	1.58	0.93–2.69	1.54	0.90–2.62

^1^ OR, odds ratio; ^2^ aOR, adjusted odds ratio: aORs for age and highest level of education were adjusted for each other only; aORs for all other variables were adjusted for age, highest level of education, and each other. * *p* < 0.05.

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
