# Peer review of "Facilitators and Barriers Associated with Uptake of HIV Self-Testing among Men Who Have Sex with Men in Chongqing, China: A Cross-Sectional Survey"

_ijerph, 2020, doi:10.3390/ijerph17051634_

Round 1

Reviewer 1 Report

Liu and colleague in their report study the factors associated with HIV self-testing in Chongqing. The study involves large number of participants with appropriate inclusion and exclusion criteria. The study design is robust. However, the data derived from the study is not novel. All these factors are already known and have been widely published in several articles. The only merit of the study is that it deals with a particular region of China. This observation does not make manuscript less important. The manuscript is well within the scope of the journal and may be accepted for the publication after vigorous typographical and grammatical checking.

Author Response

Dear Reviewers,

  We really appreciate all of your constructive comments. All of you have helped us improve the quality of this manuscript significantly. In addition, we thank the journal for giving us the second opportunity to revise this manuscript. Enclosed please find our revised manuscript where we have addressed all your comments. 

Best regards,

Mengliang Ye

February 13, 2020

Reviewer 2 Report

This is a cross-sectional study that was conducted to describe the HIV self-testing (HIVST) status of men who have sex with men (MSM) in Chongqing, China. MSM participants were recruited by random sampling, and qualified interviewers collected data by using self-administered questionnaires. The proportion of HIVST uptake was 15.6% (470/3.054), most of them ranged in age from 18 years to 36 years, 87% were unmarried, and the majority self-identified as MSM.

The accuracy of self-reported HIVST kit results was assessed by comparison with conventional kits using HIV antibody detection results. HIVST was associated with low sensitivity. The consistency between HIVST kits and antibody detection results was 90.5% (95%, CI, 87.5%-93.0%), and the kappa value showed a substantial agreement. However, it is difficult to interpret these results once in the methodology section of the manuscript, there is no description of the HIVST kits that were used, and whether they have been validated in-country, and as well do not describe which enzyme-linked immunosorbent assays (ELISA) were used for the first screening. In this context, it is difficult to confirm the findings of accuracy and consistency assessed by comparison with ELISA HIV kits.

Also, the study has several limitations already addressed by the authors, mainly due to the cross-sectional nature of the study. In conclusion, the study revealed that young age and infrequent condom use facilitated HIVST, while self-reported existing barriers for HIVST included lower educational levels and lower scores in basic HIV/AIDS knowledge. The authors address that the government should formulate guidelines to ensure the quality of self-test kits made available to this group. Unfortunately, detailed and more rigorous research methods are needed to confirm these results.

Author Response

Dear Reviewer,

  We really appreciate all of your constructive comments. All of you have helped us improve the quality of this manuscript significantly. In addition, we thank the journal for giving us the second opportunity to revise this manuscript. Enclosed please find our revised manuscript where we have addressed all your comments.

Best regards,

Mengliang Ye

February 13, 2020

Author Response

(The authors gave the same response as above.)

Round 2

Reviewer 3 Report

Thank you for your revisions. I am happy with the revised article.